# Non-Invasive Assessment of Metabolic Dysfunction-Associated Steatotic Liver Disease in Adolescents with Severe Obesity; Patient Characteristics and Association with Leptin—A Cross-Sectional Overview of Baseline Data from a RCT

**DOI:** 10.3390/children11080965

**Published:** 2024-08-11

**Authors:** Fien de Boom, Ali Talib, Yvonne G. M. Roebroek, Givan F. Paulus, Ger H. Koek, Simon G. F. Robben, Bjorn Winkens, Nicole D. Bouvy, L. W. Ernest van Heurn

**Affiliations:** 1Department of Surgery, Maastricht University Medical Center, 6229 HX Maastricht, The Netherlands; fien.deboom@maastrichtuniversity.nl (F.d.B.);; 2NUTRIM School for Nutrition and Translational Research in Metabolism, Maastricht University, 6211 LK Maastricht, The Netherlands; 3Department of Gastroenterology and Hepatology, Maastricht University Medical Center, 6229 HX Maastricht, The Netherlands; 4Department of Radiology, Maastricht University Medical Center, 6229 HX Maastricht, The Netherlands; 5Department of Methodology and Statistics, CAPHRI Care and Public Health Research Institute, Maastricht University, 6200 MD Maastricht, The Netherlands; 6Department of Pediatric Surgery, Amsterdam University Medical Centers, 1105 AZ Amsterdam, The Netherlands

**Keywords:** severely obese, obesity, adolescents, MASLD, prevalence, associated factors, leptin

## Abstract

**Background:** Metabolic dysfunction-associated steatotic liver disease (MASLD) is the most common liver disease in adolescents with obesity. Nonetheless, the guidelines for screening and managing this disease are contradictory. The purpose of this study is to non-invasively assess the prevalence, patient characteristics, and potential associated factors of MASLD in this demographic. **Methods:** This study used baseline data from an RCT in adolescents eligible for bariatric surgery. MASLD was determined by measuring the hepatorenal index (HRI) with ultrasonography, blood liver tests, and the fatty liver index (FLI). **Results:** A total of 56 adolescents enrolled in the analyses; the majority were female *n* = 44 (79%); the mean age was 15.75 (±1.01) years; the average body mass index (BMI) was 44.08 (±5.16) kg/m^2^. In 35 (62.5%) of the patients, the HRI was abnormal. This group had a higher waist/hip ratio, elevated liver biochemistry, and significantly lower leptin levels (mean difference = −46.35, 95% CI −76.72, −15.99) compared to the group with a normal HRI. In 32 (55.2%) of the patients, ALT levels were elevated and the FLI was abnormal in all (100%) participants. Linear regression analysis indicated associations between the HRI and typical anthropometric and metabolic measurements; and an inverse association between HRI and leptin B = −0.003 (95% CI −0.005, −0.00), independently of sex- and age-adjusted BMI. **Conclusions:** MASLD is highly prevalent in adolescents with severe obesity. However, the prevalence strongly depends on which tool is used, namely the HRI (62.5%), alanine transaminase levels (55.2%), and FLI (100%). Our findings suggest that leptin may be a valuable biomarker in supporting the diagnosis of MASLD.

## 1. Introduction

The prevalence of overweight and obesity in children and adolescents is high and is still rising in middle- and low-income countries [1,2] It is associated with obesity-related comorbidities, including metabolic dysfunction-associated steatotic liver disease (MASLD). MASLD, formerly known as non-alcoholic fatty liver disease (NAFLD) [3,4], is an umbrella term for elevated fat accumulation in the liver associated with metabolic disturbances, such as glucose abnormalities, hypertension, elevated plasma triglycerides, and decreased high-density lipoprotein (HDL). It is associated with the development and aggravation of extrahepatic diseases like type 2 Diabetes Mellitus (T2DM), cardiovascular diseases, and cancer [5,6,7,8]. In children, MASLD is now the most common liver disease, which emphasizes the importance of assessing patient characteristics and potential associated factors of this disease.

MASLD, like other obesity-associated diseases, has several causes, with insulin resistance as the important driving factor [9]. This obesity-associated comorbidity is of great concern, as it may develop into cirrhosis and lead to end-stage liver disease in adulthood [5]. Screening for MASLD in adolescents with obesity is challenging and much heterogeneity in diagnostic tools is observed in the literature, causing difficulties in estimating the prevalence in this demographic. Guidelines from the Society for Pediatric Gastroenterology, Hepatology and Nutrition (NASPGHAN and ESPGHAN), recommend screening all children with obesity (age > 9 years old) for MASLD by testing alanine amino transaminase (ALT) levels, even though doing so has substantial limitations [10,11]. Regarding imaging, abdominal ultrasonography is recommended by the European guidelines but the North American guidelines recommend against it because of its suboptimal diagnostic value. Some authors consider the fatty liver index (FLI) as a non-invasive screening tool for MASLD. However, this prediction tool has not been widely validated as a marker for MASLD in adolescents with obesity [12]. Guidelines and previous studies are inconsistent with regard to the estimated prevalence and screening tools for MASLD in adolescents with obesity and this is a limitation in the current literature, which may result in underdiagnosis and undermanagement of this disease [13,14].

This prospective study aims to assess the prevalence, patient characteristics, and associated factors of MASLD in a well-characterized group of adolescents with severe obesity, using the hepatorenal index (HRI), ALT levels, and the FLI. Although not as accurate as the gold standards (liver biopsy and magnetic resonance imaging), these non-invasive tools (NITs) offer non-invasive time- and cost-effective tools for screening for MASLD. Using highly sensitive cut-off values for these NITs is purported to facilitate more descriptive and precise diagnosis of patients. Ultimately, our goal is to contribute to the establishment of a consensus regarding the decision to screen adolescents eligible for bariatric surgery, as well as identifying associated factors that can support the diagnosis.

## 2. Materials and Methods

### 2.1. Participants

This study used the baseline data from a prospective randomized controlled trial in adolescents eligible for bariatric surgery. The study’s rationale, design, and applied methodology were reported in the study protocol (the Bariatric Surgery in Children (BASIC) trial, NCT01172899) [15].

### 2.2. Inclusion Criteria

Information regarding the inclusion and exclusion criteria was published previously [15]. In short, it concerned adolescents with severe obesity (BMI ≥ 40 or ≥35 with the presence of at least one obesity-associated comorbidity). The exclusion criteria were inadequate social support from family or guardians, eating disorders, severe cardiorespiratory impairment, skeletal or developmental immaturity, and syndromic disorders causing obesity. Informed consent was obtained from all participants and their parents or guardians.

### 2.3. Diagnostic Methods

Due to the invasive, time-intensive, and costly nature of the gold standards for diagnosing MASLD (liver biopsy or magnetic resonance imaging), we opted for multiple NITs to determine the prevalence of MASLD. Ultrasonography of the liver was performed by one experienced pediatric radiologist to determine the HRI (Box 1). Sporadically, in the case of unavailability of this radiologist, another experienced radiologist was consulted. A highly sensitive cut-off value of ≥1.28 was used to identify liver steatosis [16].

Box 1Formula for the HRI.HRI = Echogenicity of the liver/echogenicity of the kidney


The FLI was calculated using an algorithm based on triglycerides, BMI, gamma-glutamyl transferase (GGT), and waist circumference (WC) [17]. (Box 2) The cut-off value for hepatosteatosis was an FLI ≥ 60.


Box 2Algorithm used to calculate FLI.FLI = y/(1 + y) × 100y = 0.953 × ln (triglycerides, mg/dL) + 0.139 × BMI, kg/m^2^
+ 0.718 × ln (GGT, U/L) + 0.053 × waist circumference (cm) − 15.745

ALT levels were measured and highly sensitive cut-off values of 25 U/L for boys and 22 U/L for girls were applied.

To measure the risk of advanced fibrosis in this population non-invasively, the index for liver fibrosis (FIB-4) and the NAFLD fibrosis score (NFS) were used. (Box 3) Cut-off values for these non-invasive tools are only established for adults and are, respectively, 1.3 and −0.1413 and 1.257 [18].

Box 3Formulas for FIB-4 score and NFS.FIB-4 = Age (years) × AST (IU/L)/platelet count
(×109/L) × √ALT (IU/L).NFS = −1.675 + 0.037 × age (year) + 0.094 × BMI (kg/m²) +
1.13 × DM (with = 1, without = 0) + 0.99 × AST/ALT ratio − 0.013 × platelets
(×109/L) − 0.66 × Alb (g/dL).

### 2.4. Study Outcome Measurements

The primary outcome measurement involved assessment of the prevalence of MASLD and the patients’ characteristics in this homogeneous and well-characterized population of adolescents with severe obesity. Prevalence was assessed through the HRI with a cut-off value of ≥1.28, ALT levels with a cut-off value of >22 U/L for girls and >25 U/L for boys, or the FLI with a cut-off value of ≥60. In addition, this study assessed factors potentially associated with MASLD, using the HRI as a surrogate for MASLD.

### 2.5. Statistical Analyses

Normally distributed numerical variables are presented as mean ± standard deviation (SD), while categorical data are presented as the number of adolescents and percentage. To compare demographic and clinical variables between patients with an abnormal HRI to patients with normal HRI, an independent samples *t*-test was used for numerical variables. Multivariable linear regression analyses were used to evaluate the association between MASLD (using HRI as a surrogate marker) and potentially relevant factors, including anthropometric variables, glycemic factors, lipid profile, and leptin. The variables were assessed separately in the model, due to the sample size. We corrected for BMI z-score in the regression analyses, given its known significant role as a confounder. Assumptions were checked using graphical methods, including scatterplots for linearity, Q-Q-plots and histograms for normality, and residual plots for homoscedasticity. To check for selection bias, we compared the group that entered analysis with the group that did not due to missing data using independent samples *t*-tests. IBM SPSS Statistics for Windows (version 27.0; Armonk, NY, USA) was used for the aforementioned statistical analyses. A two-sided *p*-value ≤ 0.05 was considered statistically significant.

## 3. Results

### 3.1. Participants

A total of 59 participants were included in the trial; 3 (5.1%) did not receive abdominal ultrasonography (US) at baseline, due to the unavailability of a pediatric radiologist or missed appointments. The baseline characteristics of the remaining 56 participants are presented in Table 1. The mean age of the (mainly female) population (78.6%) was 15.75 (±1.01). Anthropometric measurements included BMI, mean 44.08 (±5.16); BMI z-score, 3.53 (±0.29); and WC, 126.96 (±13.70). The mean ALT, AST, and GGT were 30.52 (±15.26), 25.96 (±10.72), and 28.87 (±16.12), respectively. The mean HRI was 1.56 (±0.47), and FIB-4 and NFS-scores were normal in all the participants. The group that entered analysis (*n* = 56) did not significantly differ from the group that did not (*n* = 59).

#### 3.1.1. Prevalence and Patient Characteristics

MASLD was detected in 35 (62.5%) participants with an ultrasonographically calculated HRI. In 32 (55.2%) participants, ALT levels were above the cut-off value, and all (100%) patients had an abnormal FLI (Figure 1). Table 2 shows the comparison of patients with normal versus abnormal HRI. Participants with an abnormal HRI had a significantly higher waist/hip ratio (mean difference = 0.06, 95% CI 0.01, 0.11), higher ALT (mean difference = 9.97, 95% CI 1.89, 18.05), and higher AST (mean difference = 6.11, 95% CI 9.88, 9.80) levels compared to their counterparts. Interestingly, leptin was significantly lower in the group, with an HRI above the cut-off level (mean difference = −46.35, 95% CI −76.72, −15.99). The patients with abnormal ALT levels had a significantly higher weight (mean difference = −12.53, 95% CI −22.92, −2.15), waist circumference (mean difference = −8.88, 95% CI −15.98, −1.78), and HRI (mean difference = −0.30, 95% CI −0.54, −0.06). AST levels were abnormal in nine (16.1%) of the participants and those patients had a significantly higher weight (mean difference = −18.38, 95% CI −32.36, −4.41), higher BMI (mean difference = −4.62, 95% CI −8.20, −1.03), higher HRI (mean difference = −0.36, 95% CI −0.69, −0.03), and higher waist circumference (mean difference = −8.52–15.71, −1.34) A total of 10 (21.7%) participants had abnormal GGT levels, but no significant patient characteristics were observed in this group.

#### 3.1.2. Associated Factors

Multivariable linear regression analyses indicated significant associations between HRI and WC (unstandardized regression coefficient (B)) = 0.02, 95% CI 0.00, 0.03), waist/hip ratio (B = 1.58, 95% CI 0.19, 2.98), AST (B = 0.02, 95% CI 0.00, 0.03), homeostatic model assessment for insulin resistance (HOMA-IR) (B = 0.03, 95% CI 0.005, 0.06), and Leptin (B = −0.003, 95% CI −0.005, 0.00), after correction for BMI-z-score (Table 3).

## 4. Discussion

This study identified an alarmingly high prevalence of MASLD among adolescents with severe obesity. The prevalence varied strongly upon which diagnostic tool was used, namely the HRI (62.5%), ALT levels (55.2%), and FLI (100%). The HRI was used as a surrogate marker for MASLD, alongside the identification of metabolic and anthropometric factors associated with MASLD, and an inverse association with leptin was observed.

### 4.1. Prevalence of MASLD

We found an estimated prevalence of 55.2–100% of MASLD in this group of adolescents with severe obesity. A recent review identified a prevalence of MASLD in European children and adolescents with obesity of 9.5–36.4% [19,20]. The higher prevalence of MASLD in our study is probably due to the extreme obesity of our patients (mean BMI 44.08 ± 5.16 kg/m^2^). Furthermore, much heterogeneity in diagnostic tools and participants is observed in the literature, resulting in great variability in prevalence. Xanthakos et al. conducted research with a study population comparable to ours and reported a similar prevalence (59%) of MASLD obtained by liver biopsy (gold standard) [21]. The risk of fibrosis was low in our patients, as all participants had an FIB-4 and NFS below the cut-off value. This concurs with previous studies, in which liver fibrosis and liver failure in this young population were found to be less common, arguably due to the relatively short exposure to metabolic dysregulation at this age [20,21].

### 4.2. Validity of NITs and Discrepancies in Prevalence of MASLD

The demand for non-invasive accurate screening tests for MASLD in adolescents with obesity is high. In particular, there is a need for highly sensitive tools that can exclude MASLD in high-risk groups. This allows clinicians to enhance patient selection for additional costly and invasive tests. By implementing highly sensitive cut-off values for our screening tools, we excluded MASLD in 37.5% of the study population when using the HRI (cut-off value ≥ 1.28). The diagnostic quality of the HRI varies from poor to good in the literature but is likely the result of methodological heterogeneity, namely varying the use of (non-gold standard) index tests, such as the FLI and fibroelastography [22,23,24,25]. However, when its validity is tested against a gold index test (liver biopsy), a sensitivity of 100% is achieved with a cut-off value of ≥1.28. Notably, this test is not validated in adolescents; as a result, we applied the adult reference value.

This study lowered the cut-off value for ALT levels to 25 U/L for boys and 22 U/L for girls, deviating from the 50 U/L commonly used in the literature. It has been suggested that this cut-off level (50 U/L) is set too high for screening for MASLD in adolescents and patients can therefore be missed. Lowering the threshold improves sensitivity to 80% in boys and 92% for girls, while specificity remains roughly unchanged, yet is still suboptimal for screening this high-risk group [26].

The FLI revealed higher prevalence of MASLD than the HRI and ALT levels. The FLI algorithm uses raw BMI and WC, rather than z-scores, and lacks pediatric reference ranges. This may result in insufficient accuracy for predicting MASLD in this particular group. In aiming to exclude MASLD in this high-risk population, a sensitivity of >95% (≤5% false negatives) is desired, a level of accuracy that is not met by the FLI. One study compared the FLI in an obese pediatric population with magnetic resonance imaging (gold standard), and it showed only moderate diagnostic accuracy when predicting MASLD in adolescents with severe obesity; thus, FLI is not validated in this group of patients [27]. Supporting this hypothesis, the FLI remains roughly unchanged when adjusting for GGT or ALT in the formula, given a BMI and WC as high as in our study population; a ceiling effect may be occurring.

### 4.3. Patient Characteristics and Associated Factors

An inverse association between leptin and HRI was found (independently of BMI-z score). In line with this finding, the group with an HRI above the cut-off value (≥1.28) had significantly lower leptin levels. Leptin, a hormone derived from adipocytes, is hypothesized to play a role in MASLD. The etiology underlying this complex interplay is yet to be fully understood but may lie in the anti-steatotic effect of leptin and its role in glucose and lipid metabolism [28]. To our knowledge, only two studies have looked into the role of leptin in MASLD in the pediatric population and the results were contradictory. One study including 72 children with biopsy proven MASLD found hyperleptinemia to be associated with an increased risk of developing MASLD and leptin levels increased as steatosis was more severe [29]. However, this concept has been recently challenged by a cross-sectional study of 97 pre-pubertal children with obesity, wherein lower leptin z-scores were identified in patients diagnosed with MASLD through ultrasound [30]. A meta-analysis [31] including adults found a significant mean standarized difference in circulating leptin levels between patients with MASLD with fibrosis and patients without MASLD. Patients with MASLD and fibrosis had higher leptin levels than those without MASLD. Thus, the literature is contradictory, which may be explained by differences in the severity of MASLD. It can be hypothesized that the anti-steatotic effect of leptin only applies in the early stages of MASLD, but if the disease progresses, leptin could exert proinflammatory and profibogenic actions [32,33]. Since our study included adolescents with low fibrosis scores, who had been exposed to obesity for a shorter duration compared to adults, MASLD was likely in an early stage. Therefore, patients with higher circulating leptin levels had a possible beneficial effect on steatosis.

In line with earlier research [14,34], typical metabolic and anthropometric factors were significantly associated with HRI (which was used as surrogate for MASLD). BMI and BMI z-score were not significantly associated with MASLD, whereas WC and waist/hip ratio did show an association, suggesting that they are more effective for predicting MASLD [35,36]. This supports the notion behind the adoption of the new nomenclature, characterizing MASLD as a consequence of the metabolic syndrome rather than primary hepatic pathology.

### 4.4. Strengths and Limitations

A unique feature of this study is the homogenous well-characterized group that concerned only adolescents with severe obesity eligible for bariatric surgery. Although a unique feature, the use of a selective cohort that excludes normal-weight controls and consists of patients with the most severe cases of obesity introduces selection bias, limiting the generalizability to the broader population. Due to the invasive nature of liver biopsy and the unavailability of MRI, multiple NITs were used to screen for MASLD. By implementing highly sensitive cut-off values, satisfactory diagnostic accuracy was created for screening for MASLD with the HRI. Nevertheless, some limitations of the NITs need to be acknowledged. Regarding US, its accuracy can be influenced by increased subcutaneous adipose tissue, which may conceal the liver and reduce the precision of the assessment. Additionally, US is less sensitive at detecting mild steatosis and it may be challenging to differentiate between severities of hepatosteatosis. Also, it is important to consider that ALT is a marker of liver injury rather than a direct indicator of MASLD. Elevated ALT levels can suggest MASLD, but normal ALT levels do not necessarily exclude the presence of MASLD. Moreover, ALT levels can fluctuate and might not consistently reflect the degree of MASLD. Assessment of MASLD at multiple time points could decrease the variability and is a limitation of the cross-sectional study design. These limitations notwithstanding, we used a combination of NITs to reduce the potential sources of bias. This study encountered some missing data due to missing appointments, or unavailability of a pediatric radiologist. However, given the reasons for the missing data, these factors are assumed to be completely random. Furthermore, the study population consists mainly of females (78%). According to the literature, at least in adolescents, MASLD is more common in males than females [37]. However, due to the sample size of the study and existing correction for the BMI z-scores, no additional correction for sex was made in the main analysis. In addition to leptin levels, measurement of adiponectin may have been valuable because of its role in MASLD pathogenesis. Unfortunately, we did not have the means to obtain this measurement.

To conclude, the estimated prevalence of MASLD was alarmingly high but varied strongly based on the diagnostic tool used. This high prevalence emphasizes the importance of screening adolescents with severe obesity eligible for bariatric surgery for this disease. However, there is a need for novel non-invasive prediction models that can screen more accurately for MASLD in the obese pediatric population. Perhaps leptin could be used in this model, as it was associated with MASLD when HRI served as a surrogate. A longitudinal study investigating causal relationships between metabolic factors and MASLD while correcting for confounders is needed. This would be highly valuable for identifying new biomarkers for predicting MASLD in adolescents. Until then, several NITs will be helpful for predicting MASLD in patients eligible for bariatric surgery. Strong evidence of the superiority of one NIT for predicting MASLD in adolescents is lacking. Therefore, using a combination of NITs, such as ALT levels, US, and FibroScan, is recommended.

## Figures and Tables

**Figure 1 children-11-00965-f001:**
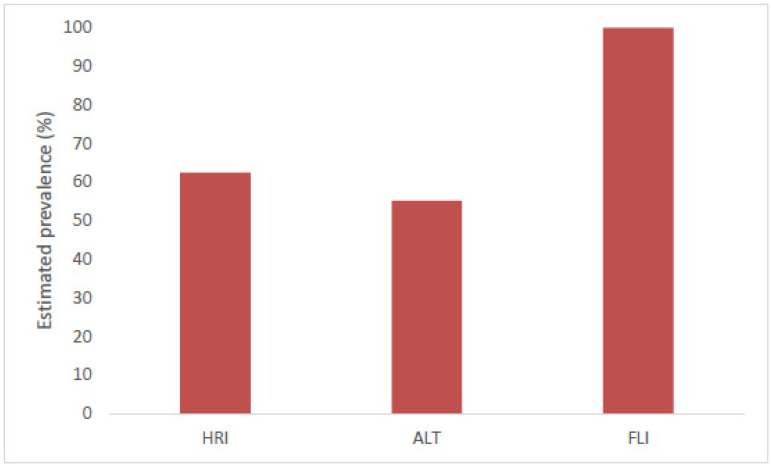
Estimated prevalence of MASLD in this study population, measured through HRI, ALT levels, and FLI. HRI—hepatorenal index; ALT—alanine transaminase; FLI—fatty liver index. Cut-off values for HRI were ≥ 1.28; cut-off values for ALT levels were 25 U/L for boys and 22 U/L for girls; cut-off values for FLI were ≥ 60.

**Table 1 children-11-00965-t001:** Baseline characteristics for the participants who entered analysis.

Variable	Total *n* = 56 Mean (±SD)	No (%) of Patients with Abnormal Laboratory Values	Missing *n* (%)
Age (years)	15.75 (±1.01)	-	-
Sex *n* (% female)	44 (79%)	-	-
BMI (kg/m^2^)	44.08 (±5.16)	-	-
BMI z-score	3.53 (±0.29)	-	-
Waist circumference (cm)	126.96 (±13.70)	-	1 (1.8%)
Waist/length ratio	0.72 (±0.19)	-	-
Waist/hip ratio	0.98 (±0.09)		-
ALT (U/L)	30.52 (±15.26)	32 (55.2%)	-
AST (U/L)	25.96 (±10.72)	9 (16.1%)	-
GGT (U/L)	28.87 (±16.12)	10 (21.7%)	10 (17.9%)
HRI	1.56 (±0.47)	35 (62.5%)	-
FLI	96.30 (±5.10)	44 (100%)	12 (21.4%)
Total cholesterol (mmol/L)	4.61 (±0.86)	20 (35.7%)	-
HDL-C (mmol/L)	1.12 (±0.34)	21 (37.5%)	-
LDL-C (mmol/L)	2.71 (±0.84)	17 (30.4%)	-
TC/HDL	4.48 (±1.57)	-	-
Triglycerides (mg/dL)	1.80 (±0.88)	23 (41.1%)	-
HOMA-IR	7.12 (±4.87)	-	1 (1.8%)
Leptin (Ug/L)	78.56 (±41.11)	50 (100%)	6 (10.7%)
FIB-4 score	0.23 (±0.05)	0 (0%)	6 (10.7%)
NFS-score	−2.39 (±1.42)	0 (0%)	-

Data presented as mean (±SD). BMI—body mass index; BMI z-score—body mass index adjusted for age and sex; FLI—fatty liver index; HRI—hepatorenal index; AST—aspartate transaminase; ALT—alanine transaminase; GGT—gamma-glutamyl transferase; TC—total cholesterol; TG—triglycerides; HDL—high-density lipoprotein; LDL—low-density lipoprotein; HOMA-IR—homeostasis model assessment-estimated insulin resistance; FIB-4—fibrosis score-4; NFS—the non-alcoholic fatty liver disease fibrosis score. Cut-off values used: AST (females 12–33 U/L, males 14–39 U/L), ALT (females 22 U/L, males 25 U/L), GGT (females < 35 U/L, males < 36 U/L), TC (<5 mmol/L), HDL (>1 mmol/L), HDL, LDL (<3 mmol/L), TG (<2 mmol/L), leptin (females 2.1–24.2 Ug/L, males 0.3–7.7 Ug/L).

**Table 2 children-11-00965-t002:** Characteristics of the participants with and without MASLD when HRI is used as a screening tool. Observed differences between the MASLD and non-MASLD group with the *p*-value are presented.

Variable	*n*	HRI ≥ 1.28 Mean (±SD)	*n*	HRI ≤ 1.28 Mean (±SD)	Mean Difference (95% CI)	*p*-Value
Age (years)	35	15.77 (±0.94)	21	15.71 (±1.14)	0.06 (−0.51, 0.62)	0.84
BMI (kg/m^2^)	35	43.85 (±5.59)	21	44.47 (±4.45)	−0.63 (−3.50, 2.25)	0.66
BMI z-score	35	3.53 (±0.32)	21	3.53 (±0.22)	−0.006 (−0.17, 0.15)	0.94
Waist circumference (cm)	34	129.60 (±13.93)	21	122.69 (±12.47)	6.91 (−0.55, 14.37)	0.07
Waist/length ratio	35	0.71 (±0.23)	21	0.74 (±0.08)	−0.02(−0.13, 0.08)	0.65
Waist/hip ratio	35	1.00 (±0.09)	21	0.94 (±0.08)	0.06 (0.01, 0.11)	**0.01 **
ALT (mmol/L)	35	34.26 (±15.62)	21	24.29 (±12.67)	9.97 (1.89, 18.05)	**0.02 **
AST (mmol/L)	35	28.26 (±11.71)	21	22.14 (±7.60)	6.11 (0.37, 11.86)	**0.04 **
GGT (mmol/L)	27	28.30 (±15.64)	18	28.33 (±16.61)	−0.04 (−9.88, 9.80)	1.00
FLI	26	97.52 (±2.73)	18	94.54 (±7.01)	2.99 (−0.07, 6.04)	0.06
Total cholesterol (mmol/L)	35	4.64 (±0.96)	21	4.56 (±0.71)	0.08 (−0.40, 0.56)	0.74
HDL (mmol/L)	35	1.09 (±0.35)	21	1.16 (±0.33)	−0.07 (−0.26, 0.12)	0.47
LDL (mmol/L)	35	2.71 (±0.92)	21	2.70 (±0.71)	0.01 (−0.46, 0.48)	0.96
Triglycerides (mg/dL)	35	1.95 (±0.97)	21	1.54 (±0.67)	0.41 (−0.07, 0.89)	0.09
HOMA-IR	34	7.50 (±5.38)	21	6.52 (±3.96)	0.98 (−1.75, 3.70)	0.48
Leptin (Ug/L)	34	51.02 (±53.41)	21	97.37 (±53.32)	−46.35 (−76.72, −15.99)	**0.003**

Data presented as mean (±SD), mean differences with 95% confidence interval, and a two-sided *p*-value ≤ 0.05 was considered statistically significant. A cut-off value of HRI ≥ 1.28 was applied. The bold highlights statistical significance. BMI—body mass index; BMI z-score—body mass index adjusted for age and sex; FLI—fatty liver index; HRI—hepatorenal index; AST—aspartate transaminase; ALT—alanine transaminase; GGT—gamma-glutamyl transferase; TG—triglycerides; HDL—high-density lipoprotein; LDL—low-density lipoprotein; HOMA-IR—homeostasis model assessment-estimated insulin resistance.

**Table 3 children-11-00965-t003:** Linear regression analysis for HRI with potential associated factors, corrected for BMI-z-score. * not corrected.

Variable	Unstandardized Regression Coefficient (B)	95% CI	*p*-Value
BMI (kg/m^2^) *	0.002 *	−0.004, 0.008	0.49
BMI z-score *	−0.24 *	−0.68, 0.20	0.29
Waist circumference (cm)	0.02	0.00, 0.03	**0.02**
Waist/length ratio	0.04	−0.67, 0.76	0.90
Waist/hip ratio	1.58	0.19, 2.98	**0.03**
ALT (U/L)	0.008	0.00, 0.02	0.06
AST (U/L)	0.02	0.003, 0.03	**0.01**
GGT (U/L)	0.01	−0.003, 0.02	0.18
FLI *	0.02 *	−0.01, 0.05	0.23
Total cholesterol (mmol/L)	−0.003	−0.15, 0.15	0.96
HDL (mmol/L)	−0.23	−0.62, 0.16	0.24
LDL (mmol/L)	−0.02	−0.17, 0.14	0.85
Triglycerides (mg/dL)	0.14	−0.002, 0.28	0.05
HOMA-IR	0.03	0.005, 0.06	**0.02 **
Leptin (Ug/L)	−0.003	−0.005, 0.00	**0.02 **

Data presented as unstandardized regression coefficient (B) with 95% confidence intervals (CI). The bold highlights statistical significance. BMI—body mass index; BMI z-score—body mass index adjusted for age and sex; FLI—fatty liver index; HRI—hepatorenal index; AST—aspartate transaminase; ALT—alanine transaminase; GGT—gamma-glutamyl transferase; HDL—high-density lipoprotein; LDL—low-density lipoprotein; HOMA-IR—homeostasis model assessment-estimated insulin resistance.

## Data Availability

The participants of this study did not give written consent for their data to be shared publicly, so due to the sensitive nature of the research, supporting data are not available.

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
