# Peer review of "Non-Invasive Assessment of Metabolic Dysfunction-Associated Steatotic Liver Disease in Adolescents with Severe Obesity; Patient Characteristics and Association with Leptin—A Cross-Sectional Overview of Baseline Data from a RCT"

_children, 2024, doi:10.3390/children11080965_

Round 1

Reviewer 1 Report

Comments and Suggestions for Authors

The editors present an interesting manuscript regarding the prevalence of MASLD in a population of adolescents with severe obesity. The article is to the point with no unnecessary information. I only have a few proposals/questions.

1. How many patients had abnormal laboratory values (apart from ALT levels which is already mentioned); this should be added in table 1 or at least mentioned in the manuscript. Moreover, what were the  characteristics of these patients? Likewise, did you have any patients with FIB-4 or NFS score suggestive of increased fibrosis? If yes, what were their characteristics? 

2. It would be interesting to have adiponectin measurements in these patients, since adiponectin seems to be importan in MASLD pathogenesis. If this is not possible then the fact that they also measured leptin should be added in limitations section.

3. The authors should propose which non-invasive biomarker is, in their opinion, the most helpful for predicting MASLD in these patients until a newer biomarker, like leptin, is found 

Comments on the Quality of English Language

Quality of English fine; minor editing required

Reviewer 2 Report

Comments and Suggestions for Authors

I have read de Boom et al.'s study entitled "Non-invasive assessment of Metabolic dysfunction associated steatotic  liver disease in adolescents with severe obesity; patient  characteristics and association with leptin". Please find below my comments for revision. These comments aim to improve the study's scientific rigor, transparency, and comprehensive understanding of the findings:

1. You should mention your study type in the Title.

2. In the Methods, the authors should justify the selection of specific diagnostic tools and cut-off values for the non-invasive assessment of MASLD in adolescents with severe obesity.

3. Regarding the "Diagnostic methods" section, it is important to address the limitations or potential sources of bias associated with the non-invasive tools used to assess MASLD (e.g., discussing the potential impact of confounding factors or the accuracy of the diagnostic tests in this specific population).

4. In the "Statistical analyses", the authors should provide detailed information on the selection of statistical methods and the rationale behind the choice of adjustment variables in the regression analysis.

5. The authors should acknowledge additional limitations of the study, such as the cross-sectional design, potential selection bias, and the need for further longitudinal investigations to establish causal relationships between metabolic factors and MASLD in adolescents with severe obesity.

Round 2

Reviewer 1 Report

Comments and Suggestions for Authors

The authors have adequately answered my concerns and the manuscript is now, as far as I am concerned, suitable for publication

Comments on the Quality of English Language

Minor editing required

Reviewer 2 Report

Comments and Suggestions for Authors

Thanks for your revisions.